



# Measurement Report: Changes of ammonia emissions since the 18th century in south-eastern Europe inferred from an Elbrus (Caucasus, Russia) ice core record

Michel Legrand[1,2], Mstislav Vorobyev[3], Daria Bokuchava[3], Stanislav Kutuzov[4,5], Andreas Plach[5], Andreas Stohl[6], Alexandra Khairedinova[3], Vladimir Mikhalenko[3], Maria Vinogradova[3], Sabine Eckhardt[7], and Susanne Preunkert[1]

[1]Observatoire des Sciences de l'Univers de Grenoble, Université Grenoble Alpes, CNRS, Grenoble, 38400, France
[2]Université Paris Cité and Université Paris Est Creteil, CNRS, LISA, Paris, F-75013 France
[3]Institute of Geography, Russian Academy of Sciences, Moscow, 119017, Russia
[4]School of Earth Sciences, The Ohio State University, Columbus, OH, 43210, USA
[5]Byrd Polar and Climate Research Center, Columbus, OH, 43210, USA
[6]Department of Meteorology and Geophysics, University of Vienna, Vienna, 1010, Austria
[7]Department of Atmospheric and Climate Research, NILU - Norwegian Institute for Air Research, Lillestrøm, N-2027, Norway

*Correspondence to*: Mstislav Vorobyev (mslavavo@gmail.com)

**Abstract.** To investigate the historical levels of atmospheric ammonia ($NH_3$) pollution in south-eastern Europe, a 182 m long ice core was extracted from Mount Elbrus in the Caucasus, Russia. This ice core contains a record of ammonium ($NH_4^+$) levels from ~1750 CE (Common Era) to 2009 CE. The $NH_4^+$ ice core record indicates a 3.5-fold increase of annual concentrations from $34 \pm 7$ ng g$^{-1}$ (~1750-1830) to $117 \pm 23$ ng g$^{-1}$ over the recent decades (1980-2009). The increase remained moderate until 1950 CE (mean concentration of $49 \pm 14$ ng g$^{-1}$ over the 1830-1950 period), and then accelerated to reach a maximum close to 120 ng g$^{-1}$ in 1989. This ice core trend is compared to estimated past anthropogenic $NH_3$ emissions in Europe by using state-of-the-art atmospheric transport modeling of submicron aerosols (FLEXPART model driven with 0.5°x0.5° ERA5 reanalysis data). It is shown that in summer, when both vertical atmospheric mixing and agricultural $NH_3$ emissions are strengthened, the $NH_4^+$ ice core trend is in good agreement with the course of estimated $NH_3$ emissions from south-eastern Europe since ~1750 with a main contribution from south European Russia, Turkey, Georgia, and Ukraine. Examination of Mount Elbrus ice deposited over the second half of the 18th century when agricultural activities were less than 10% of those during the 1990s, suggest a pre-1750 annual $NH_4^+$ ice concentration related to natural emissions of 25 ng g$^{-1}$. This pre-1750 natural level mainly related to natural soil emissions represents ~20% of the 1980-2009 $NH_4^+$ level, a level mainly related to current agricultural emissions that almost completely outweigh biogenic emissions from natural soils.



# 1 Introduction

Gaseous ammonia is the most abundant alkaline gas in the atmosphere and represents a major component of total reactive nitrogen. It plays an important role in determining the overall acidity (alkalinity) of precipitation. A large portion of atmospheric aerosols, acting as cloud condensation nuclei, consists of sulfate neutralized to various extents by $NH_3$. Ammonia and ammonium (collectively abbreviated as $NH_x$) are key nutrients that fertilize plants. Too large inputs of N to the environment may, however, lead to eutrophication of terrestrial and aquatic ecosystems and thus threaten the biodiversity (Asman et al., 1998; Galloway et al., 2003). Therefore, the growing $NH_3$ emissions resulting from fertilization applied to meet the need to sustain food production for a growing human population impact the environment. Consequently, it is important to have a clear understanding of the $NH_3$ sources. Natural sources include wildfires especially at high northern latitudes, natural soils, and ocean whereas anthropogenic emissions are dominated by agriculture including animal husbandry and $NH_3$-derived fertilizer application (Galloway et al., 2004). The temporal change of anthropogenic emissions needs to be accurately documented since increasing temperature resulting from climate change, may amplify the $NH_3$ emissions from soils to the atmosphere (Skjøth and Geels, 2013; Sutton et al., 2013), counteracting expected benefits from emission control measures (Simpson et al., 2014). The uncertainties of natural/biogenic emissions over continents are much larger than those associated with anthropogenic sources, partly due to the complexity and variability of natural ecosystems. Furthermore, $NH_3$ present in seawater mainly comes from the biological decomposition of organic matter by bacteria (Johnson et al., 2007), however, there is still an ongoing debate regarding the significance of this natural source in the global atmospheric budget (Paulot et al., 2015). Past $NH_4^+$ aerosol concentration trends extracted from ice cores contain key information on past growth of anthropogenic $NH_3$ emissions as well as the partitioning between natural and anthropogenic emissions at regional scales. Such studies remain scarce, and up to now only two studies have compared ammonium trends extracted from Alpine ice cores to atmospheric chemistry-transport models (Engardt et al., 2017; Fagerli et al., 2007) with the aim to constrain past ammonia emissions in western Europe over the 20[th] century.

Here we present a seasonally resolved ice core record of $NH_4^+$ deposition extracted from a 182 m long ice core drilled in 2009 at Mount Elbrus in the Caucasus. Our main goal was to assess the magnitude of natural $NH_3$ sources in south-eastern Europe and the importance of anthropogenic emissions since ~1750. This was accomplished by comparing the ice core records to estimated past anthropogenic $NH_3$ emissions and using state-of-the-art FLEXPART atmospheric transport and deposition simulations of aerosol.

## 2 Materials and methods

### 2.1. Ice core dating and analyses

A deep ice core was drilled to bedrock (182.6 m) in 2009 on the western plateau of Mount Elbrus (ELB, 43°N, 42°E; 5115 m above sea level, asl) in the Caucasus (Russia). The upper 168.6 m (131.5 meters water equivalent, mwe) depth of the ice core





were first dated by annual layer counting using pronounced seasonal variations in ammonium and succinate concentrations, both exhibiting well-marked winter minima (Mikhalenko et al., 2015; Preunkert et al., 2019). The annual counting was found to be very accurate (± 1yr) over the last hundred years when anchored with the stratigraphy of the 1912 Katmai horizon located at 116.7 m (87.7 mwe) depth (Mikhalenko et al., 2015). As a result of glacier ice flow, the annual ice layer thickness decreases

with depth, also rendering the dating increasingly uncertain with depth. Though dating uncertainties prior to 1912 were not quantified in these two previous studies, an age of 1774 CE was assigned for the 168.6 m depth layer (Preunkert et al., 2019), the ice layer corresponding to the large eruption of Tambora (1815 CE) being, however, not identified. Based on complementary data including the acidity, the dating was recently revisited by Mikhalenko et al. (2024), suggesting the presence of the 1815 Tambora horizon either at 153.7 or 154.7 m depth and an age of CE 1752 ± 4 years at 168.6 m depth.

Whereas back to 1890 CE, no significant difference appeared between the two dating estimates, a difference by ~25 years is observed at the end of the record.

Ice cores were subsampled and decontaminated at -15°C using an electric plane tool detailed in Preunkert and Legrand (2013). In brief, ice samples were first cut with a band saw, and all surfaces of the cut samples were decontaminated by removing ~3 mm with a pre-cleaned electric plane tool under a clean air bench. Due to the glacier ice flow, annual layer thickness decreases

from 1.5 mwe near the surface to 0.18 mwe at 157 m (i.e., 122 mwe) depth. As detailed in Preunkert et al. (2019), a total of 3724 subsamples were obtained along the upper 168.6 m of the core, and to minimize the loss of temporal resolution with depth, the sample depth resolution was decreased from 10 cm at the top to 5 cm at 70 m (47 mwe) and 2 cm at 157 m (121.8 mwe) depth and below. With that, we still sampled on average 10-12 samples per year at 157 m depth (compared to 25-30 samples per year near the surface), rendering identification of winter layers possible down to 168.6 m depth.

Chemical measurements were done with a Dionex ICS-1000 chromatograph equipped with a CS12 separator column for cations ($Na^+$, $K^+$, $Mg^{2+}$, $Ca^{2+}$, and $NH_4^+$), a Dionex 600 equipped with an AS11 separator column for anions ($Cl^-$, $NO_3^-$, and $SO_4^{2-}$) and light carboxylates. Detailed working conditions are given in Legrand et al. (2013). For all investigated ions, blanks of the ice decontamination procedure were found to be insignificant with respect to the respective levels found in the ice cores.

**2.2 FLEXPART model simulations**

The $NH_4^+$ trends in ELB ice will be examined with respect to estimates of past anthropogenic $NH_3$ emissions from south-eastern Europe. Present in the gas phase, $NH_3$ is rapidly converted into submicron aerosol, preferentially reacting with $H_2SO_4$ to produce stable ammonium salts (($NH_4)_2SO_4$ and/or $NH_4HSO_4$) rather than with $HNO_3$ to produce semi-volatile $NH_4NO_3$. Having longer atmospheric lifetimes than $NH_3$ (owing to removal by precipitation of a few days to a week instead of less than one day for $NH_3$), ammonium salts are transported and deposited at larger distances from source regions (Van Pul et al., 2009).

Ideally, comparing observed $NH_4^+$ ice concentrations (or deposition fluxes) with past $NH_3$ emissions would require simulations made with a chemistry-transport model that accounts for the conversion of gaseous $NH_3$ into $NH_4^+$ aerosol, and transport/deposition of $NH_x$ from countries located around the drill site. The increase of acidity following the growth of $SO_2$





and NO$_2$ emissions in western Europe was shown to increase the formation of NH$_4^+$ aerosol permitting transport of NH$_x$ over longer distances and inducing a non-linearity between NH$_3$ emissions in source regions and NH$_4^+$ deposition over the Alps

(Fagerli et al., 2007). The situation in the Caucasus is even more complex due to a large increase of dust emissions over the recent decades (Kutuzov et al., 2019; Preunkert et al., 2019) that may have counterbalanced the effect of growing SO$_2$ and NO$_2$ on the NH$_x$ partitioning (see discussions in Sect. 5.1). Finally, most chemistry transport models dealing with past NH$_3$ emissions did not consider the effect of climate fluctuations on the volatility of NH$_3$ emitted from soils.

A fully coupled chemistry transport model with accurate pH calculations that considers changes of SO$_2$ and NO$_2$ as well as

dust, and potential impact of climatic conditions on the volatility of NH$_3$ is presently not available. We here used a simpler approach that only accounts for effect of atmospheric transport and deposition of submicron sulfate aerosol without considering the NH$_x$ partitioning (assuming that a large fraction of NH$_3$ is rapidly converted into submicronic NH$_4^+$ aerosol). To do that, we used backward simulations of the state-of-the-art Lagrangian particle dispersion model FLEXPART (FLEXible PARTicle dispersion model; Eckhardt et al., 2017) that determines the sensitivity of deposition at the ELB site to submicron aerosol

emissions in Europe. The model was run at monthly intervals for the 1980-2019 years, and particles were traced backward from the observation site for 30 days. Simulations were done with the recent ERA-5 reanalysis at a resolution of 0.5°x0.5° (137 vertical layers, of which 41 are located below 5000 m asl; Hersbach et al., 2020), the ELB grid point being located at a model elevation of 2400 m. Since the model topography at the ELB location is substantially lower than the real altitude of the ELB site, deposition fluxes simulated by FLEXPART are somewhat ambiguous. We chose to simulate deposition both at the

model surface at the ELB location as well as by accounting for wet deposition only above the real height of ELB, i.e., removing all simulated deposition below the real ELB altitude. Figure 1 shows averaged emission sensitivities for the ELB site, representing a source-receptor relationship that maps the sensitivity of deposition at the site (receptor) to an emission flux (source).



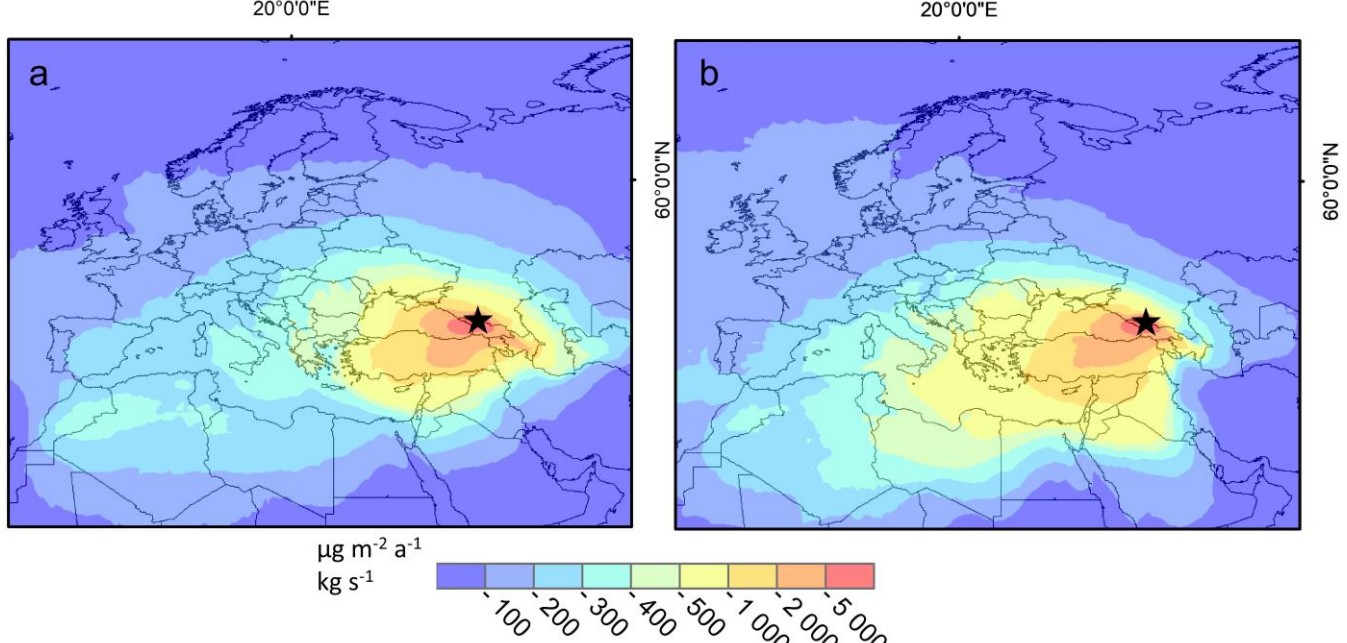

**Figure 1. Emissions sensitivities in ((mg m$^{-2}$ yr$^{-1}$)/kg s$^{-1}$) at the ELB ice-core site (black star) based on FLEXPART model simulations of sulfate aerosol transport and deposition for summer (a) and winter (b) (real elevation).**

Past $NH_4^+$ deposition fluxes at the ELB site were calculated by weighting past $NH_3$ emissions from each grid cell of the inventory by its emission sensitivity and then summing over all grid cells to get the simulated deposition rate. Calculations were conducted with the CDO (Climate Data Operator) software. We used the global dataset of anthropogenic $NH_3$ emissions (Hoesly et al., 2018) in which data are presented in the Network Common Data Form (NetCDF) format and have 0.5°x0.5° spatial resolution. Hoesly et al. (2018) provided anthropogenic emissions from the Community Emissions Data System (CEDS) for the years 1750 to 2014 with monthly time resolution and data are divided into eight sectors, namely transportation, energy, international shipping, industrial, agriculture, waste, solvents production and application, and residential sectors. Figure 2 illustrates the change of annual gridded ammonia emissions in Europe from 1880 to 2005.




**Figure 2. Gridded ammonia anthropogenic emissions (0.5°x0.5°, Hoesly et al., 2018, https://github.com/JGCRI/CEDS/) in western and south-eastern Europe in 1880 CE, 1950 CE, 1988 CE, and 2005 CE.**

The validity of the approach that compares $NH_4^+$ ice core trends with estimates of past $NH_3$ emissions using the FLEXPART model for dispersion of submicron aerosol will be discussed in Sect. 5. However, an important assumption is that the

atmospheric transport has not changed systematically over the course of the ice core record and that the climatological emission sensitivity values obtained for the period 1980-2019 can thus be applied to the full ice core record. While this assumption may not be fully valid, it is likely that changes in the atmospheric transport climatology had a much smaller effect on simulated deposition at the ELB site than the changes in emissions over the same period. Furthermore, to evaluate the bias due to the non-consideration of $NH_3$ loss, FLEXPART simulations were also done and compared to ELB observations for deposition of

sulfate using past $SO_2$ anthropogenic emissions (Sect. 6).



### 3. Ice core data presentation

Using the winter ammonium/succinate minima we determined half-year summer and winter means from 1748 to 2009 (Fig. 3). Despite a more pronounced loss of resolution in winter layers (12 samples per winter near the surface and 1-2 samples per winter at 157 m depth), the seasonal dissection was still possible down to 168.6 m depth (Sect. 2.1), but some of the winter ammonium/succinate minima used to define a winter layer were too thin to determine reliably winter $NH_4^+$ concentrations in the part of the core. Of the 262 winters (1748-2009), 21 winters were missed along the 20 lower meters (i.e., prior to ~1850 CE) of the core. In Figure 3, we also report annual concentrations calculated as the arithmetic mean of winter and summer. When the winter half-year value was missing, we averaged values of the preceding and the following winters to calculate the arithmetic annual value. In order to minimize the effect of the interannual variability of meteorological conditions on the ice record, as it is generally observed at mountain sites (Fagerli et al., 2007), we smoothed the ice core records (first component of single spectra analysis with a 5-year time window, Figure 3).





**Figure 3. Past summer (a), winter (b), and annual (c) changes of ammonium ice concentrations. Open dots are individual values, red curves are smoothed profiles (first component of single spectra analysis with a 5-yr time window). The blue curves are the smoothed profiles when samples containing dust had been removed (Section 3).**




As discussed by Preunkert et al. (2019), the arrival at ELB of large amounts of dust material is accompanied by enhanced ice concentrations for numerous species including sulfate, nitrate as well as ammonium. Both large dust plumes originating from the Middle East and Sahara reach the Caucasus (Kutuzov et al., 2013) and dust emissions from the Levant region had changed over time in response to enhanced occurrence of droughts in North Africa/Middle East regions and soil moisture content in the

Levant regions, respectively (Kutuzov et al., 2019). The causes of such increased ammonium concentrations in samples containing large amounts of dust are not clear. On the one hand, the presence of dust may increase uptake and oxidation of $SO_2$ as well as uptake of nitric acid (Dentener et al., 1996), decreasing their availability to promote formation of $(NH_4)_2SO_4$ and/or $NH_4HSO_4$ aerosols, on the other hand it promotes uptake of $NH_3$ together with nitric acid and formation of $NH_4NO_3$ aerosol (Usher et al., 2003). In Figure 3, we examined to what extent these past changes of dust had impacted those of

ammonium by comparing the ammonium trends when samples containing large dust amounts were considered ($NH_4^+$) or not (reduced $NH_4^+$ denoted $NH_4^+$red in the following). It is seen that whatever the season, the effect of dust remained insignificant prior to 1845 but became significant essentially in summer with a mean difference between $NH_4^+$ and $NH_4^+$red of 32 ng g-1 for half-year summer values from 1985 to 2000. Since the difference remained close to 15 ng g-1 from 1900 to 1950 and 19 ng g-1 from 1950 to 2000, the dust changes did not significantly modify the overall ammonium trend. In the following, when

discussing the long-term trends of $NH_4^+$ in ELB ice with respect to past anthropogenic $NH_3$ emissions, we will consider the trend of $NH_4^+$red concentrations.

## 4. The natural ammonium level

In addition to oceanic emissions, ammonia is naturally emitted by natural soils, wild animals, and natural fires. At the global scale, wild animals and soils under natural vegetation are estimated to emit 2.5 Tg N yr-1 and 2.4 Tg N yr-1, respectively (Sutton

et al., 2013), natural fires to emit 1.6 Tg N yr-1 (Galloway et al., 2004). These natural emissions represent ~20-30% of the present-day $NH_3$ budget that is dominated by emissions from agricultural activities. Related to manure use prior to World War I, anthropogenic $NH_3$ emissions then became mainly due to the use of nitrogen fertilizers produced from $NH_3$ synthetized by the Haber Bosch process. Compared to other continents such as America, Europe is very poor in fauna in both diversity and number, weakening the contribution of natural emissions by wild animals. Natural lightning-induced forest fires in Europe are

limited compared to Siberia and Canada. Quantification of sources of ammonia from natural soils and vegetation remains a challenge with bi-directional $NH_3$ fluxes of which magnitude and direction vary with the type of ecosystem as well as management and environmental variables (Sutton et al., 2013). Whereas Buijsman et al. (1987) proposed an emission of 0.6 Tg N yr-1 from natural soils in Europe (i.e., 10% of total emissions in the early 1980s), using a more recent estimate of fluxes from natural soils, Sutton et al. (1995) derived a far lower emission. From that, setting natural soil emissions to zero, Simpson

et al. (1999) estimated that in Europe natural $NH_3$ sources only represent ~1% of the present-day budget mainly due to wild animals and forest fires. If correct and referring to $NH_4^+$ levels observed in ice recently deposited at ELB, natural sources would contribute for only 2 ppb to $NH_4^+$ concentrations of ELB ice (Fig. 3). The present-day situation with an absence of





natural soil emissions and anthropogenic sources representing 99% of total $NH_3$ emissions had changed over the past due a decrease of anthropogenic emissions and a growth of natural soils at the expense of agricultural areas. In the following,

neglecting the contribution from wild animals and forest fires, we have assumed that when agricultural activities reached a maximum at the end of the 20th century, in the quasi-absence of left unmanaged soils in Europe $NH_4^+$ concentrations in ELB ice were only due to anthropogenic activities without any contribution from natural emissions.

It is likely that the $NH_4^+$ minimum observed in ELB ice deposited during the second part of the 18th century (mean annual concentration of $34 \pm 7$ ng g$^{-1}$ from 1750 to 1800, Fig. 3c) already exceeded the natural background level in this region. Whereas

emissions of $SO_2$ and $NO_2$ in 1750 CE (i.e., well before the onset of the industrial era in Europe in ~1850 CE) represented less than 1% of recent emissions, those of $NH_3$ indeed were already 5% to 16% of their 1980-2000 values in most countries contributing to $NH_4^+$ deposition at the ELB site (Table 1). This earlier occurrence of significant pollution for $NH_3$ compared to $SO_2$ and $NO_2$ in south-eastern Europe is also clearly depicted in past emission inventories with emissions from southern Russia, for instance, reaching 10% of their 1980-2000 mean values in 1810 for $NH_3$, compared to 1913 for $SO_2$, and 1934 for

$NO_2$ (Hoesly et al., 2018).

**Table 1. Emissions of $NH_3$, $SO_2$, and $NO_2$ (CEDS inventories, Hoesly et la., 2018) from countries mainly contributing to $NH_4^+$ deposition at the CDD (France: FR, Italy: IT, Spain: ES) and ELB (Turkey: TR, Russia: RU, Southern Russia: SRU, Ukraine: UA, Georgia: GE, Romania: RO, Iran: IRN, Azerbaidjan: AZ, and Egypt: EG) sites. Δ denotes the change from the 1900-1930 years to**
**1970-1990 (1980-2000) years.**

| | 1750 CE Emission (Gg yr$^{-1}$) | 1900-30 CE Emission (Gg yr$^{-1}$) | 1970-1990 Emission (Gg yr$^{-1}$) | Δ (GMole yr$^{-1}$) | $(\Delta SO_2 + \Delta NO_2)/\Delta NH_3$ molar ratio |
|---|---|---|---|---|---|
| FR | $NH_3$: 142 | $NH_3$: 277 | $NH_3$: 760 | $NH_3$: 28.4 | 1.9 |
| | $SO_2$: 30 | $SO_2$: 668 | $SO_2$: 1870 | $SO_2$: 18.8 | |
| | $NO_2$: 18 | $NO_2$: 197 | $NO_2$: 1850 | $NO_2$: 35.9 | |
| IT | $NH_3$: 60 | $NH_3$: 192 | $NH_3$: 487 | $NH_3$: 17.4 | 3.1 |
| | $SO_2$: 3 | $SO_2$: 84 | $SO_2$: 1880 | $SO_2$: 28.1 | |
| | $NO_2$: 4 | $NO_2$: 54 | $NO_2$: 1280 | $NO_2$: 26.6 | |
| ES | $NH_3$: 36 | $NH_3$: 106 | $NH_3$: 380 | $NH_3$: 16.1 | 2.4 |
| | $SO_2$: 2.5 | $SO_2$: 188 | $SO_2$: 1530 | $SO_2$: 21.0 | |
| | $NO_2$: 1.5 | $NO_2$: 31 | $NO_2$: 809 | $NO_2$: 16.9 | |
| TR | $NH_3$: 51 | $NH_3$: 119 | $NH_3$: 530 | $NH_3$: 24.2 | 1.4 |
| | $SO_2$: 1 | $SO_2$: 17 | $SO_2$: 1340 | $SO_2$: 20.7 | |
| | $NO_2$: 3 | $NO_2$: 12 | $NO_2$: 619 | $NO_2$: 13.2 | |
| RU | $NH_3$: 123 | $NH_3$: 715 | $NH_3$: 1630 | $NH_3$: 53.8 | 5.4 |
| | $SO_2$: 18 | $SO_2$: 538 | $SO_2$: 10500 | $SO_2$: 155.6 | |





| | | | | | |
|---|---|---|---|---|---|
| | $NO_2$: 7 | $NO_2$: 227 | $NO_2$: 6430 | $NO_2$: 134.8 | |
| SRU | $NH_3$: 16 | $NH_3$: 97 | $NH_3$: 227 | $NH_3$: 7.6 | 3.4 |
| | $SO_2$: 1 | $SO_2$: 47 | $SO_2$: 838 | $SO_2$: 12.4 | |
| | $NO_2$: 1 | $NO_2$: 24 | $NO_2$: 654 | $NO_2$: 13.7 | |
| UA | $NH_3$: 47 | $NH_3$: 310 | $NH_3$: 942 | $NH_3$: 37.2 | 2.7 |
| | $SO_2$: 2 | $SO_2$: 247 | $SO_2$: 4120 | $SO_2$: 60.5 | |
| | $NO_2$: 2.5 | $NO_2$: 67 | $NO_2$: 1900 | $NO_2$: 39.8 | |
| GE | $NH_3$: 7 | $NH_3$: 19 | $NH_3$: 44 | $NH_3$: 1.5 | 2.0 |
| | $SO_2$: 0.1 | $SO_2$: 4 | $SO_2$: 80 | $SO_2$: 1.2 | |
| | $NO_2$: 0.2 | $NO_2$: 4 | $NO_2$: 87 | $NO_2$: 1.8 | |
| RO | $NH_3$: 23 | $NH_3$: 104 | $NH_3$: 338 | $NH_3$: 13.8 | 1.7 |
| | $SO_2$: 1.5 | $SO_2$: 78 | $SO_2$: 1080 | $SO_2$: 15.6 | |
| | $NO_2$: 1.5 | $NO_2$: 52 | $NO_2$: 438 | $NO_2$: 8.4 | |
| IRN | $NH_3$: 17 | $NH_3$: 53 | $NH_3$: 359 | $NH_3$: 18.0 | 2.5 |
| | $SO_2$: 0.4 | $SO_2$: 7 | $SO_2$: 1610 | $SO_2$: 25.0 | |
| | $NO_2$: 1.7 | $NO_2$: 9 | $NO_2$: 910 | $NO_2$: 19.6 | |
| AZ | $NH_3$: 4 | $NH_3$: 13 | $NH_3$: 60 | $NH_3$: 2.8 | 1.7 |
| | $SO_2$: 0.1 | $SO_2$: 2 | $SO_2$: 149 | $SO_2$: 2.3 | |
| | $NO_2$: 0.2 | $NO_2$: 2 | $NO_2$: 111 | $NO_2$: 2.4 | |
| EG | $NH_3$: 19 | $NH_3$: 43 | $NH_3$: 260 | $NH_3$: 12.8 | 0.6 |
| | $SO_2$: 1 | $SO_2$: 7 | $SO_2$: 381 | $SO_2$: 5.8 | |
| | $NO_2$: 1 | $NO_2$: 10 | $NO_2$: 87 | $NO_2$: 1.6 | |

To estimate the natural $NH_4^+$ levels we examined the relationship between ice concentrations (C) and deposition fluxes (φ) calculated by the FLEXPART model (Fig. 4a) using past $NH_3$ emissions. As seen in Figure 5a for summer, using calculated φ values at the real elevation of the ELB site, the C and φ parameters are linearly correlated with an y-intercept 39.7 ± 2.4 ng g⁻¹ that would correspond to the $NH_4^+$ concentration related to natural soil emissions before occurrence of significant agricultural $NH_3$ emissions that started with the agricultural revolution at the beginning of the 15[th] century. Using φ values at the surface elevation we obtained a similar y-intercept (C = 0.85 φ + 39.2 with $R^2$ = 0.77 instead of C = 1.5 φ + 39.7 with $R^2$ = 0.77). Only the use of $NH_4^+$red values instead of $NH_4^+$ ones lead to a slight decrease of the y-intercepts for summer (35.0 ± 2.2 ng g⁻¹, Fig. 5b).






**Figure 4. Ammonium deposition calculated by FLEXPART at the Elbrus site. (A) Deposition calculated at the real elevation (Section 2.2) for half-year summer and winter and over the year using the seasonal CEDS inventories (Hoesly et al., 2018). (B) Sources of ammonium deposition at the Elbrus site at the real elevation in summer using the summer CEDS inventories.**



**Figure 5. Relationship between summer ice concentrations of ammonium (a for NH₄⁺ and b for NH₄⁺red and deposition fluxes (φ) at ELB calculated at the real elevation in summer by FLEXPART using estimated past anthropogenic NH₃ emissions (CEDS inventories, Hoesly et al. (2018)).**





For winter, no significant difference between $NH_4^+$ and $NH_4^+red$ concentrations can be observed (Fig. 3b). In contrast to summer, the winter trend surprisingly indicates a mean value in ice deposited over the second part of the 18th century ($18 \pm 6$

ng $g^{-1}$ from 1750 to 1800, Fig. 3b) that is significantly higher than the one over the 1920-1945 time period ($12 \pm 6$ ng $g^{-1}$) with no significant difference between $NH_4^+$ and $NH_4^+red$ concentrations. Furthermore, the C and $\varphi$ parameters are less well linearly correlated for winter (C = 0.28 $\varphi$ + 14.2 with $R^2$ = 0.26, Figure 6b) than summer (Figure 5a) leading to a more uncertain estimate of the y-intercept. As discussed in Section 3, due to lowering of sampling resolution with depth of winter layers, the winter half-year values were calculated with a limited number of samples and are therefore more uncertain, particularly prior

to the middle of the 19th century (see also Fig. 6a). That may have an impact on the winter trend as suggested by the eight winter half-year values exceeding 30 ng $g^{-1}$ in ice deposited prior to 1900 (Fig. 3b and Fig. 6a) that were calculated with only one sample. Discarding winter half-year values calculated with one sample, reduced the number of winter values exceeding 25 ng $g^{-1}$ prior to 1900 CE, improved the correlation between C and $\varphi$ (C = 0.34 $\varphi$ + 11.5 with $R^2$ = 0.37) and resulted in a slightly lower y-intercept ($11.5 \pm 1.3$ ng $g^{-1}$) than when all values including those based on one sample were considered (14.2

ng $g^{-1}$). An even lower y-intercept ($8.1 \pm 1.6$ ng $g^{-1}$) is calculated when considering winter layers consisting of more than 3 samples. In spite of these uncertainties, ice core data suggest a typical winter natural concentration close to ~10 ng $g^{-1}$.

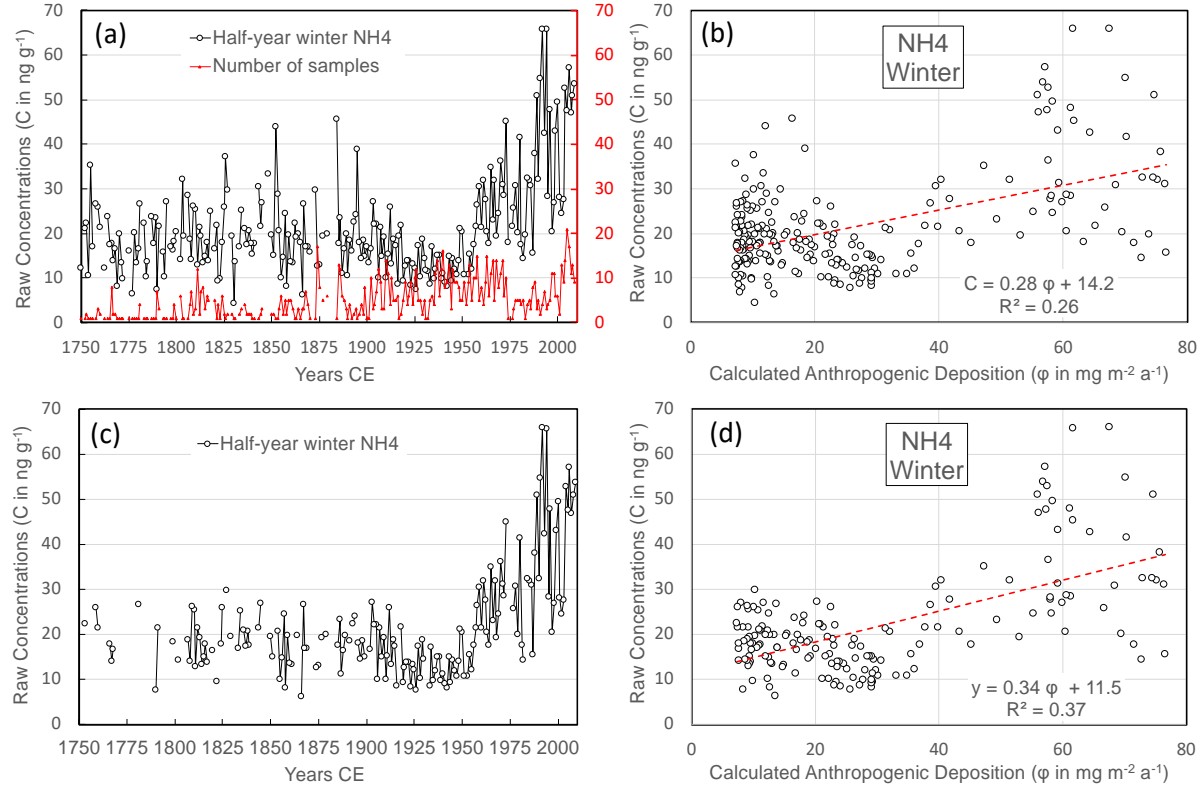

**Figure 6. (a): Winter ice-core record of ammonium concentrations (black dots), the red line indicates the number of values available to calculate individual half-year winter means. (b) Relationship between winter ice concentrations of $NH_4^+$ (C) and deposition fluxes**
**($\varphi$) at ELB calculated at the real elevation in winter by FLEXPART using estimated past anthropogenic $NH_3$ emissions (CEDS**



**inventories, Hoesly et al. (2018)). (c) and (d) are similar to (a) and (b) but discarding winter half-year means when only one sample was available.**

The preceding findings suggest a natural $NH_4^+$ level close to 37 ng g$^{-1}$ and 10 ng g$^{-1}$, in summer and winter, respectively,
leading to an annual value ~22 ng g$^{-1}$ representing ~20% of the mean 1980-2009 $NH_4^+$ value instead of 1% at the end of the 1980s.

## 5. The anthropogenic ammonium trend

In the following we focus on the summer trend and only briefly discuss the winter trend since (1) the ice core record of pollution is better documented for summer, (2) the $NH_3$ pollution is higher in summer than winter.

### 5.1 The ammonium pollution in summer

Comparison of estimated past anthropogenic emissions with changes in ice requires an accurate assessment and removal of the contribution of natural emissions to ice concentrations. Mainly related to soils under natural vegetation, for ammonia, quasi-null at the end of the 20$^{th}$ century, these emissions have changed over the past. The natural $NH_4^+$ level in 1750 CE can be estimated from the relationship between observed ice concentrations and deposition fluxes calculated by FLEXPART. For
$NH_4^+$red in summer, extrapolation of the recent values ($\varphi$ of 110 mg m$^{-2}$ yr$^{-1}$ for C of 184 ng g$^{-1}$) to the simulated deposition of 8 mg m$^{-2}$ yr$^{-1}$ in 1750 (Fig. 5b) suggests a remaining anthropogenic contribution of 14 ng g$^{-1}$ to the total $NH_4^+$red concentration of 46 ng g$^{-1}$, and thus a natural $NH_4^+$red contribution of 32 ng g$^{-1}$ in 1750.

Accurate estimates of past natural $NH_4^+$red concentrations between 1750 (32 ng g$^{-1}$) and 2009 (~0 ng g$^{-1}$) is not straightforward, using neither available data on past changes of agricultural area (and unmanaged soil area) in south-eastern Europe nor past
anthropogenic deposition simulated by FLEXPART to estimate the growth back in time of natural soil area at the expense of decreasing agricultural area. Indeed, in one hand increasing agricultural soil areas and the subsequent decrease of natural soil areas from 1750 to 2009 were not linearly correlated to growing anthropogenic emissions, with growing amounts of fertilizers applied to managed soil areas. On the other hand, countries for which FLEXPART simulations indicate a significant contribution to the $NH_4^+$ deposition in ELB ice include Russia and several Middle East countries (Turkey, Iran, Egypt, Fig.
4b) for which agricultural areas are documented since 1600 CE (https://ourworldindata.org/grapher/total-agricultural-area-over-the-long-term), data being, however, not detailed enough for weighting emissions from the different countries to the $NH_4^+$ natural level at the ELB site. Given these uncertainties, we calculated past natural soil contributions as estimated from past changes of both (1) total agricultural areas in the Middle East and Russia and (2) anthropogenic deposition at the site simulated by FLEXPART (Fig. 7a). These estimates were obtained by scaling the natural contribution to zero in 1990 and 32 ng g$^{-1}$ in
1750 for $NH_4^+$red. From that, we calculated the contribution of anthropogenic sources to past $NH_4^+$red concentrations of ELB



ice (Fig. 8), denoted anthropogenic-1 and anthropogenic-2, by subtracting to total concentrations natural soil contributions derived from agricultural areas and simulated anthropogenic deposition, respectively.

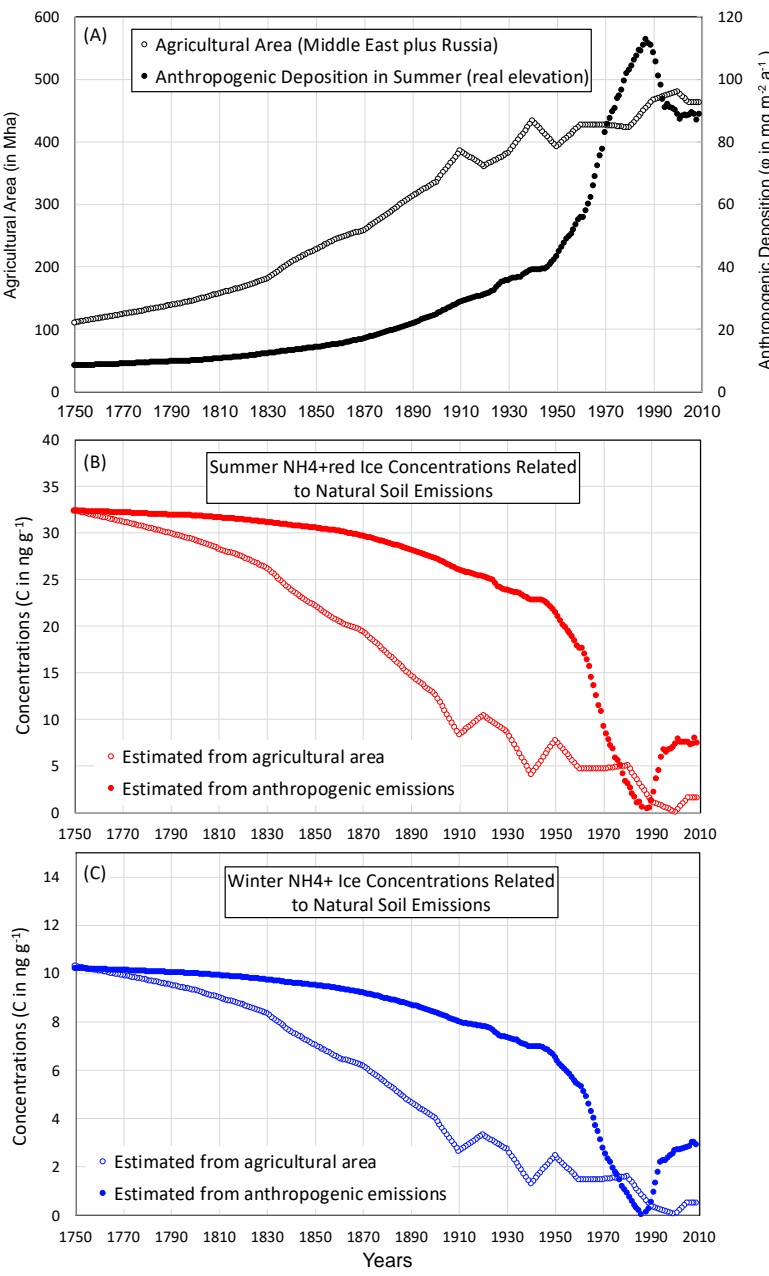

**Figure 7. Past changes of the agricultural area of the Middle East and Russia (A) (https://ourworldindata.org/grapher/total-agricultural-area-over-the-long-term) and of the contribution of ammonium concentrations in the ELB ice (B for summer, C for winter) related to emissions from natural soils (section 5). Concentrations related to natural soils emissions were estimated from changes of agricultural soils and anthropogenic emissions to calculate the anthropogenic fractions, denoted anthropogenic-1 and anthropogenic-2, respectively, in Figure 8 and 10.**





**Figure 8. (A) Summer smoothed ice-core trends of NH₄⁺red concentrations from which the natural level applying the assumptions detailed in Section 4 (denoted anthropogenic-1 and anthropogenic-2) was subtracted, and deposition fluxes at ELB calculated by FLEXPART (real and surface elevation) of ELB by using estimated past anthropogenic NH₃ emissions (CEDS inventories, Hoesly et al. (2018)). (B) Same for deposition fluxes observed in ice.**





Figure 8a compares the summer trend of $NH_4^+$red in ELB ice with deposition fluxes simulated at the real and surface elevation

at the site using past anthropogenic emissions of $NH_3$ from CEDS. The summer ice core trend is characterized by an increase

that remained limited to ~0.1% yr⁻¹ from 1750 to ~1850. The increase slightly strengthened from 1850 to 1950 (~0.2% yr⁻¹)

and became well-pronounced after ~1950 (~2.8% yr⁻¹ between 1950 and 1988). By 1988, concentrations reached a maximum

of 180 ng g⁻¹ and then decreased again in the early 1990s to reach a plateau at ~165 ng g⁻¹ until 2009. These past changes of

$NH_4^+$red ice concentrations are very consistent with past changes of deposition simulated both at the real and surface elevation

of the ELB site by FLEXPART. These deposition flux changes reflect past emission changes characterized by a growth of

$NH_3$ emissions that took place after World War II in many European countries including those that contribute to the deposition

at the ELB site (Fig. 9). Furthermore, the maximum of emissions that took place in the late 1980s, during the perestroika in

USSR, is also well recorded in ELB summer ice. As a result of decreasing $NH_3$ emissions in Russia and Ukraine in 1988-1989

(Fig. 2), Turkey became in the 1990s the main contributor of $NH_4^+$ deposition at the ELB site (Fig. 4b).

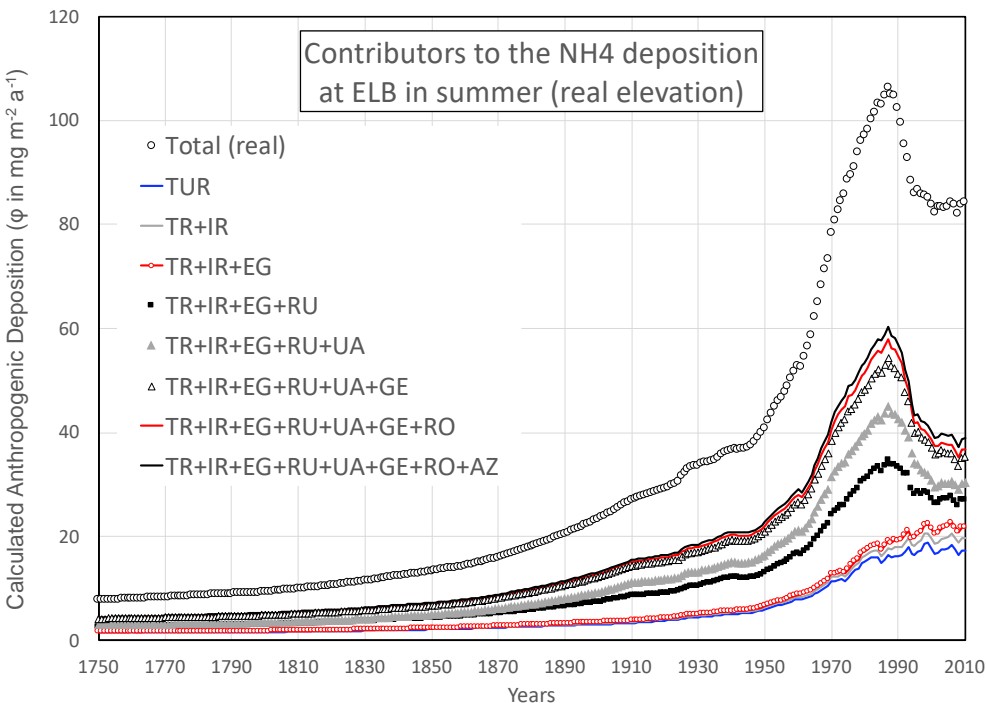

**Figure 9. Countries contributing to the ammonium deposition at the Elbrus site in summer at the real elevation using past NH₃ summer inventories (CEDS, Hoesly et al. (2018)). The solid black line refers to the total deposition. Main contributors were Turkey: TR, Russia: RU, Ukraine: UA, Georgia: GE, Romania: RO, Iran: IRN, Azerbaidjan: AZ, and Egypt: EG.**


The preceding comparison between $NH_4^+$ ice concentrations and deposition fluxes at the site simulated by FLEXPART using

estimates of past anthropogenic $NH_3$ emissions is based on the assumption that ice-concentration changes are mainly associated

with changes of depositional fluxes, being influenced both by changes of emissions and transport pathways. It also assumes

that conditions of deposition, in particular the precipitation rates at the site, did not change significantly over the time period



covered by the record. Information on past precipitation at alpine sites cannot always be derived from the ice core records because the ice core annual layer ice thickness is affected by wind-driven accumulation or erosion and therefore is not consistently representative of past precipitation. Snow redistribution by the wind at the ELB drill site was measured by stakes on the plateau during three field seasons showing, however, a zero balance between erosion and accumulation there (Mikhalenko et al., 2024) and offering the possibility to reconstruct past accumulation rates over the last 260 years. It was

shown that whereas summer precipitation fluctuated around $0.8 \pm 0.5$ mwe, no significant trend was detected from the middle of the $18^{th}$ century to the beginning of the $21^{st}$ century. It is therefore not surprising that we observed a rather similar temporal trend for observed ice concentration (Fig. 8a) and observed deposition fluxes calculated by multiplying ice concentrations by reconstructed summer precipitation rates (Fig. 8b). On average, observed deposition fluxes ranged between depositions calculated by FLEXPART at the model surface (2400 m asl) and the real elevation of the site (5115 m asl), the deposition at

the real elevation being almost only half of that at the model surface elevation. The fact that there is a good agreement does not imply that FLEXPART accurately simulates the transport and deposition of submicron aerosols to the ELB site, since FLEXPART is neglecting the immediate loss of $NH_3$ after emission. This point will be further discussed in Section 6. However, it is expected that for an otherwise perfect model, the observed deposition should fall in between the two modelled values, since they represent the extreme range of values plausible for the model when considering the differences between real and

model topography.

As already seen in Fig. 5, past $NH_4^+$ ice concentrations (or calculated deposition fluxes) are linearly correlated to deposition fluxes at the site simulated by FLEXPART. This finding is interesting since FLEXPART calculations consider past changes of $NH_3$ emissions but no changes in the NHx speciation or transport pathways. This linearity between $NH_4^+$ deposition and $NH_3$ emissions differs from what was seen at the CDD Alpine site (Fagerli et al., 2007), where the ice record showed increasing

$NH_4^+$ concentrations in summer by a factor 3 whereas $NH_3$ emissions from countries that contribute to deposition at CDD increase only by a factor of two. Such a higher enhancement of $NH_4^+$ levels in Alpine ice than in the $NH_3$ emission input suggested a higher rate of $NH_4^+$ aerosol formation over the recent decades due to a larger availability of sulfuric and nitric acid resulting from $SO_2$ and $NO_x$ emissions that have increased more than $NH_3$ emissions up to 1980 in western Europe. As seen in Table 1, though the molar $NH_3/(SO_2+NO_2)$ ratio of emissions in some countries impacting ELB than those impacting CDD,

the difference is relatively weak and may not be enough to explain the difference between the two ice records. That is confirmed by ice data reported in Table 2 showing recent changes of $NH_4^+$, $SO_4^{2-}$, and $NO_3^-$ between the two sites that remained similar. In contrast, Table 2 indicates a larger increase of calcium over the recent decades at ELB than at CDD. As a result, it seems plausible that the larger increase of alkaline dust material at ELB compared to CDD had partly neutralized the effect of acidic compounds there. This net effect of changes of acidic species and dust is thus an increase of acidity at CDD but not at ELB.

This may have led to a change of the NHx partitioning in summer in the Alps favoring the formation of $NH_4^+$ over the recent decades but not in the Caucasus where the formation of $NH_4^+$ remained unchanged over the two last centuries.



**Table 2. Concentrations of major ions and acidity in the Alpine (CDD) and Caucasus (ELB) ice deposited from 1900 to 1930 and during the ammonium maximum (1970-1990 CE at CDD, 1980-2000 CE at ELB). Δ refers to the mean changes from 1900-1930 CE to the time period of ammonium maximum. Negative concentrations of H⁺ reflect alkaline samples.**

| CDD | 1900-1930 CE | 1970-1990 CE | Δ |
|---|---|---|---|
| $SO_4^{2-}$ | $170 \pm 73$ ng g$^{-1}$ | $793 \pm 208$ ng g$^{-1}$ | $13.0$ µEq L$^{-1}$ |
| $NO_3^-$ | $93 \pm 38$ ng g$^{-1}$ | $340 \pm 97$ ng g$^{-1}$ | $4.0$ µEq L$^{-1}$ |
| $NH_4^+$ | $58 \pm 31$ ng g$^{-1}$ | $165 \pm 49$ ng g$^{-1}$ | $6.0$ µEq L$^{-1}$ |
| $Ca^{2+}$ | $49 \pm 36$ ng g$^{-1}$ | $172 \pm 112$ ng g$^{-1}$ | $6.2$ µEq L$^{-1}$ |
| $H^+$ | $+ 1.25 \pm 1.2$ µEq L$^{-1}$ | $+ 6.05 \pm 4.2$ µEq L$^{-1}$ | $+ 4.8$ µEq L$^{-1}$ |
| ELB | 1900-1930 | 1980-2000 | Δ |
| $SO_4^{2-}$ | $175 \pm 78$ ng g$^{-1}$ | $727 \pm 155$ ng g$^{-1}$ | $11.5$ µEq L$^{-1}$ |
| $NO_3^-$ | $139 \pm 33$ ng g$^{-1}$ | $420 \pm 90$ ng g$^{-1}$ | $4.5$ µEq L$^{-1}$ |
| $NH_4^+$ | $88 \pm 27$ ng g$^{-1}$ | $200 \pm 44$ ng g$^{-1}$ | $6.2$ µEq L$^{-1}$ |
| $Ca^{2+}$ | $127 \pm 62$ ng g$^{-1}$ | $401 \pm 218$ ng g$^{-1}$ | $13.7$ µEq L$^{-1}$ |
| $H^+$ | $- 2.6 \pm 2.7$ µEq L$^{-1}$ | $- 5.9 \pm 8.9$ µEq L$^{-1}$ | $- 3.3$ µEq L$^{-1}$ |

### 5.2 Ammonium pollution in winter

As discussed in Section 4, the ammonium ELB ice core trends are less well documented in winter layers. Prior to 1900, the number of samples available to calculated mean winter half-years is often limited (Fig. 6), rendering inaccurate estimates of mean winter half-year concentrations. This lack of documentation of winter layers also concerns the reconstruction of past winter accumulation rates (Mikhalenko et al., 2024) and therefore in the following we restricted our discussion of winter trends to the 1900-2009 years. Over the 20[th] century, winter concentrations are on average 1/5[th] the summer values (Fig. 3). Deposition fluxes calculated by FLEXPART from 1900 to 2009 are 40% and 30% lower in winter than in summer at the surface and real elevation, respectively (Fig. 4a). The decrease of simulated deposition fluxes from summer to winter is mainly due to higher $NH_3$ emissions in May and to a lesser extent in September than over the rest of the year in source regions impacting the ELB site (Fig. S1).

As done for summer (Fig. 7b), the observed deposition fluxes calculated by multiplying winter concentrations by reconstructed winter accumulation rates ($0.45 \pm 0.27$ mwe instead of $0.86 \pm 0.36$ mwe in summer; Mikhalenko et al. (2024)) were compared to deposition fluxes calculated by FLEXPART at the surface and the real elevation of the site over the 20[th] century (Fig. 10). Whereas, consistent with observations in ice, a marked post-1950 increase is simulated by FLEXPART, in contrast to what was observed in summer, the course of post-1950 changes poorly matched between ice observations and FLEXPART simulations. Furthermore, simulated φ FLEXPART deposition fluxes are far larger than those observed in ice (Fig. 10b). For instance, between 1950 and 2009, observed deposition fluxes in ice were some 4 and 6 times lower than simulated deposition



at real and surface elevation, respectively. Such a discrepancy may have several causes including an overestimation of NH$_3$

355    emissions in winter, an incorrect FLEXPART simulations of inversion layers that are frequent in winter, or an underestimation of the observed winter deposition due to winter snow being blown away at the ELB site. These questions will be further discussed in the following section.





**Figure 10. (A) Winter smoothed ice-core trends of NH$_4^+$ concentrations from which the natural level applying the assumptions detailed in Section 4 (denoted anthropogenic-1 and anthropogenic-2) was subtracted, and deposition fluxes at ELB calculated by FLEXPART (real and surface elevation) of ELB by using estimated past anthropogenic NH$_3$ emissions (CEDS inventories, Hoesly et al. (2018)). (B) Same for deposition fluxes observed in ice.**

## 6. Comparison between observations and FLEXPART simulations for sulfate

To evaluate the bias in simulated deposition fluxes of ammonium from FLEXPART due to the non-consideration of NH$_3$ loss, we tested the FLEXPART model for sulfate deposition using SO$_2$ emissions that, similar to NH$_3$ emissions, would overestimate the sulfate deposition. Although for both species we expect an overestimate of simulated deposition fluxes, historical SO$_2$ emissions are better documented than those of NH$_3$ (Hoesly et al., 2018). The better knowledge of anthropogenic emissions for SO$_2$ than NH$_3$ may also help to discuss further the large difference between observed and simulated NH$_4^+$ deposition in winter (Sect. 5.2). Seasonally resolved ice core ELB trends were already available for sulfate (Preunkert et al., 2019) and were recently compared to state of the art of chemistry-transport model (Earth System Model (ESM); Moseid et al., 2022).

As done for NH$_4^+$, we compare trends of observed anthropogenic concentrations and deposition of SO$_4^{2-}$red with past deposition calculated by FLEXPART using SO$_2$ inventories. In summer, concentration and deposition of SO$_4^{2-}$red remained quasi-unchanged from 1750 to ~1900 CE, increased modestly between 1900 and 1950. The increase then accelerated until ~the middle of the 1980s followed by a strong decrease from the early 1990s to 2009 (Fig. 11). The observed well-marked 1980-1992 maxima are very consistent with the maximum of deposition simulated by FLEXPART using past SO$_2$ emissions from south-eastern European countries. The maximum of sulfur pollution in these regions occurred slightly later than in western Europe and was also impacted by consequences of the perestroika at the end of the 1980s.



**Figure 11. (A) Summer smoothed ice-core trends of SO$_4^{2-}$red concentrations from which the natural level (90 ng g$^{-1}$, section 6) was subtracted, and deposition fluxes at ELB calculated by FLEXPART (real and surface elevation) of ELB by using estimated past anthropogenic SO$_2$ emissions (CEDS inventories, Hoesly et al. (2018)). (B) Same for deposition fluxes observed in ice.**



385

Despite differences in the timing, from 1750 to 2009, the observed $SO_4^{2-}$red and $NH_4^+$red deposition fluxes generally remained close to the FLEXPART-simulated depositions calculated at the real and model elevation of the site. More precisely, considering first the recent 1980-2009 time period, that overlaps the 1980-2019 years of the FLEXPART simulations, the observed deposition of 470 mg m$^{-2}$ a$^{-1}$ lied between simulated depositions at the model (610 mg m$^{-2}$ a$^{-1}$) and real (340 mg m$^{-2}$ a$^{-1}$) elevation. A similar finding is observed for $NH_4^+$red deposition fluxes, with observed value of 154 mg m$^{-2}$ a$^{-1}$ compared to 178 and 98 mg m$^{-2}$ a$^{-1}$ from simulations. From ~1950 and 1980, observed depositions were closer to deposition simulated at the real elevation, both for ammonium and sulfate. This similar decrease in observed deposition with respect to the 1980-2009 simulations would suggest a decrease of atmospheric transport towards the site by the middle of the 20$^{th}$ century compared to the recent decades. This comparison between ammonium and sulfate over the recent decades suggests that that losses of $NH_3$ and $SO_2$ that may bias the FLEXPART results are of similar magnitude.

Finally, at the beginning of the 20$^{th}$ century, simulated depositions were higher than observations for $NH_4^+$red but not $SO_4^{2-}$red. The larger uncertainties in calculating anthropogenic $NH_4^+$red deposition over the middle of the 20$^{th}$ century (Fig. 8) did not seem large enough to explain this finding. That suggest that either $SO_2$ emissions in south-eastern Europe were slightly underestimated or more likely those of $NH_3$ were overestimated.

400 In winter, $SO_4^{2-}$red observed deposition fluxes in ice deposited between 1950 and 2009 were some 5 and 7.5 times lower than simulated depositions at real and surface elevation, respectively (Fig. 12). The difference between observed and simulated depositions are similar for sulfate and ammonium and we can therefore rule out that $NH_3$ emissions in winter were overestimated. Rather, these differences may be a result of deficiencies in the FLEXPART simulations, e.g., related to an under-representation of winter-time atmospheric inversion layers in the meteorological input data which would hinder vertical transport, or an underestimation of the observed winter deposition due to winter snow being blown away at the ELB site. An incorrect FLEXPART simulations of inversion layers that are frequent in winter, or an underestimation of the observed winter deposition due to winter snow being blown away at the ELB site could be the causes of this common sulfate and ammonium discrepancy between observations and simulations. Field analysis of summer and winter accumulation at the ELB site have shown that whereas in summer snow accumulation is evenly distributed over the plateau (Mikhalenko et al., 2024), mainly due to the local topography a relocation of winter snow was observed with a net loss in the southern and western parts of the plateau and a net accumulation on the northern and eastern parts of the plateau. Although the drill site is located between these two areas, a loss of winter rendering unrepresentative winter ammonium means as well as an underestimation of winter snow accumulation (and observed deposition) is possible. Further long-term field observations based on stakes measurements for instance, would permit to better evaluate this effect.





**Figure 12. (A) Winter smoothed ice-core trends of SO₄²⁻ concentrations from which the natural level (50 ng g⁻¹, section 6) was subtracted, and deposition fluxes at ELB calculated by FLEXPART (real and surface elevation) of ELB by using estimated past anthropogenic SO₂ emissions (CEDS inventories, Hoesly et al. (2018)). (B) Same for deposition fluxes observed in ice.**



## 7. Conclusions

A record of ammonium covering the 1750-2008 years was extracted from a 182 m long ice core drilled in 2009 at Mount Elbrus in the Caucasus, Russia. That permitted to investigate ammonia pollution in south-eastern Europe well before the onset of the industrial period in 1850. The $NH_4^+$ ice core record indicates a 3.5-fold increase of concentrations, from $34 \pm 7$ ng g$^{-1}$

between 1750 and 1830, $49 \pm 14$ ng g$^{-1}$ between 1830 and 1950, to a maximum close to 120 ng g$^{-1}$ in 1989 followed by a plateau at ~115 ng g$^{-1}$ until 2009. Simulations with the FLEXPART atmospheric transport of submicron aerosol using estimated past anthropogenic $NH_3$ emissions in Europe are in good agreement with ice core data and indicated main contributions from south European Russia, Turkey, Georgia, and Ukraine. Although FLEXPART simulations did not consider changes of the $NH_x$ partitioning over the past, we find that, in contrast to what was observed in western Europe, the observed

$NH_4^+$ deposition was linearly related to simulated $NH_4^+$ deposition, and thus, $NH_3$ emissions. That suggests that in these regions the recent increase of atmospheric dust had counteracted the effect of growing emissions of $SO_2$ and NO, limiting enhanced formation and long-range transport of ammonium salts. Transport-chemistry model simulations are here welcome to further evaluate $NH_4^+$ ice core records but they would require consideration of growing $SO_2$ and NO emissions as well as dust aerosol and its heterogeneous interaction with acidic species and $NH_3$. Another interesting finding is the $NH_4^+$ level of 35 ng g$^{-1}$ in

1750 that is close to the natural level of 25 ng g$^{-1}$. Representing some 20% of the 1980-2009 level, this natural level indicates a significant contribution of natural sources to the $NH_3$ budget, contrasting with present-day conditions when agricultural activities strongly outweigh biogenic emissions from natural soils in Europe.

## Data availability

Ammonium concentrations data are available at https://zenodo.org/records/12549687 (Legrand et al., 2024).

## 440 Author contributions

The paper was written by ML, MV and SK with contributions from AP, AS, SE, SP, VM, MaV and AK. The ice core chemistry records were produced by SP and ML. MV and DB calculated aerosols deposition fluxes. AP, AS and SE calculated sensitivity by FLEXPART. All the authors read and discussed the manuscript and contributed to improving the final paper.

## Competing interests

The contact author has declared that none of the authors has any competing interests.



**Acknowledgements**

The study was completed in the laboratory created within Megagrant project (agreement no. 075-15-2021-599; 8 June 2021) with the support of the FMWS-2024-0004 project.

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
