# Peer review of "Measurement Report: Changes of ammonia emissions since the 18th century in south-eastern Europe inferred from an Elbrus (Caucasus, Russia) ice core record"

_EGUsphere, 2024_

## Author Comment (AC1)

**Reviewer 1.**

This study investigated that the historical levels of atmospheric ammonia (NH$_3$) pollution in south-eastern Europe. The results showed that The NH$^{4+}$ ice core record indicates a 3.5-fold increase of annual concentrations from 34 ± 7 ng g$^{-1}$(~1750-1830) to 117 ± 23 ng g$^{-1}$ over the recent decades (1980-2009). And this pre-1750 natural level mainly related to natural soil emissions represents ~20% of the 1980-2009 NH$^{4+}$ level, a level mainly related to current agricultural emissions that almost completely outweigh biogenic emissions from natural soils. I recommend the manuscript be revised before being accepted for publication.

*We would like to sincerely thank the reviewer for the thoughtful comments, which have helped us improve and clarify the manuscript.*

Line 50, the authors mention: "up to now only two studies have compared ammonium trends extracted from Alpine ice cores to atmospheric chemistry-transport models (Engardt et al., 2017; Fagerli et al., 2007)", but what are the key findings in these previous studies? A brief introduction to previous studies can better highlight the research content.

*These two studies were limited to past changes having occurred over the 20$^{th}$ century whereas anthropogenic NH$_3$ emissions were already significant during the 19$^{th}$ century. Also, the two studies were conducted at glaciers essentially impacted by western European emissions. In the revised version we added:*

*"Such studies remain scarce, and up to now only two studies have compared ammonium trends extracted from Alpine ice cores to atmospheric chemistry-transport models (Engardt et al., 2017; Fagerli et al., 2007) with the aim to constrain past ammonia emissions in western Europe over the 20$^{th}$ century. The short lifetime of atmospheric aerosols (days to weeks) and the regional character of ammonia emissions, however, motivated further studies conducted in other regions and extending back to the 19$^{th}$ century when agricultural activities started to be significant."*

Line 70, "a difference by ~25 years is observed at the end of the record". Does this difference have an impact on the comparison of results from analyzing observations and simulations?

*We recognize that, as it reads, the text was misleading. First, we now clarify that in this study we used the most recent dating that was established with more information than the previous one. Whereas there is indeed a difference of 25 years between the two dating around 1750, that does not represent a dating uncertainty. Instead, as now better explained in the text, the uncertainty on the depth of the Tambora layer (153.7 m or 154.7 m depth) leads to an uncertainty of 4 years around 1815 for the new dating. As discussed in section 5.1, the ammonium concentration increase remained limited to 0.1% yr$^{-1}$ between 1750 and 1850 compared to 2.8% yr$^{-1}$ after 1950, implying that dating uncertainty in the bottom part of the record would not significantly impact the discussion of the temporal trend of the last 250 years. The text has been reworded as :*

*"Based on complementary data including the acidity, the dating was recently revisited by Mikhalenko et al. (2024), suggesting the presence of the 1815 Tambora horizon either at 153.7 or 154.7 m depth and an age of CE 1752 ±4 years at 168.6 m depth. This more accurate dating was used in this study, the uncertainty around 1815 CE being of 4 years. Note that as discussed in section 5.1, the ammonium concentration increase remained limited between 1750 and 1850 compared to the post 1950 period, implying that dating uncertainty in the bottom part*

*of the record does not significantly modify discussions on the main changes that had occurred over the two last centuries."*

Lines 73-74, "removing ~3 mm with a pre-cleaned electric plane tool under a clean air bench.", What is the scientific basis for it, please add.

*This is a rather standard decontamination technique described and justified before including the paper cited already in the text (Preunkert and Legrand; 2013). We do not think that further explanation is necessary for this.*

*That ensures that the outer part of piece of ice (often contaminated) was removed permitting to obtain free-contamination piece of ice, as previously successfully tested for Greenland (Fischer et al., 1998), Alpine (Preunkert et al., 2001; Preunkert and Legrand; 2013), and Caucasus ice (Preunkert et al., 2019).*

*Fischer, H. , Wagenbach, D. and Kipfstuhl, S. (1998): Sulfate and nitrate firn concentrations on the Greenland Ice Sheet 1. Large-scale geographical deposition changes, Journal of Geophysical Research D17, 103 , pp. 21927-21934*

*Preunkert, S., M. Legrand, D. Wagenbach, and H. Fischer, Sulfate trends in a Col du Dôme (French Alps) ice core : A record of anthropogenic sulfate levels in the European mid-troposphere over the 20th century, J. Geophys. Res., 106, 31,991-32,004, 2001.*

Line 259, "countries for which FLEXPART simulations indicate a significant contribution to the NH$^{4+}$ deposition in ELB ice include Russia and several Middle East countries (Turkey, Iran, Egypt, Fig.4b)". How this "significant contribution" is judged, the graph shows that the rest of the countries are relatively high compared to Iran or Egypt.

*Taken into account, first we reworded this sentence*

*"On the other hand, countries for which FLEXPART simulations indicate a significant contribution to the NH$_4^+$ deposition in ELB ice include the former USSR (Russia, Ukraine, and Georgia) and the Middle East (Turkey) (Fig. 4b) for which agricultural areas are documented since 1600 CE (https://ourworldindata.org/grapher/total-agricultural-area-over-the-long-term), data being, however, not detailed enough for weighting emissions from the different countries to the NH$_4^+$ natural level at the ELB site."*

*Second, concerning other countries we added in the next paragraph: "These deposition flux changes reflect past emission changes characterized by a growth of NH$_3$ emissions that took place after World War II in many countries with major contributions from Russia, Turkey, Georgia, and Ukraine. As seen in Fig 4b, even with weakened emission sensitivities (Fig. 1), due to large NH$_3$ emissions (Fig. 2), other countries located further west such as Bulgaria, Albania, Hungary, Macedonia, part of Italy, Hungary, Slovakia, and Czech Republic still significantly contribute to the deposition at the ELB site. Furthermore, the maximum of emissions that took place in the late 1980s, during the perestroika in USSR, is also well recorded in ELB summer ice. As a result of decreasing NH$_3$ emissions in Russia and Ukraine in 1988-1989 (Fig. 2), Turkey became in the 1990s the main contributor of NH$_4^+$ deposition at the ELB site (Fig. 4b)."*

Fig.6b and Fig.6d have a low correlation for the scatter fit (0.26 and 0.37), can the authors try a segmented fit, which is negatively correlated up to the first half of the x-axis as can be seen in the figure.

Taken into account. We do not think that this will help here, the scattering being too high, likely due to low winter values together with a poor representativeness of the thin winter ice layers that characterized the bottom part of the ice core which is stated in the text.

Lines 374-376, "increased modestly between 1900 and 1950". That's too vague a descriptionare. It would have been clearer if a comparison in terms of data had been given. This makes the paper more rigorous and academic. Note other similar descriptions in the manuscript.

Taken into account, we now specify "In summer, concentration and deposition of $SO_4^{2-}$red remained quasi-unchanged from 1750 to ~1900 CE, increased modestly at a rate of 3.5% $yr^{-1}$ between 1900 and 1950. The increase then accelerated until ~the middle of the 1980s (12% $yr^{-1}$) followed by a strong decrease from the early 1990s to 2009 (11% $yr^{-1}$, Fig. 11)."

Lines 400-415, "In winter, $SO4^{2-}$red observed deposition fluxes in ice deposited … would permit to better evaluate this effect". The authors explain a lot about this phenomenon, but there doesn't seem to be a clear explanation, the authors should summarize and analyze to get a clear point of explanation.

Taken into account.

It seems to us that it was stated in the draft as "*The difference between observed and simulated depositions are similar for sulfate and ammonium and we can therefore rule out that NH₃ emissions in winter were overestimated. Rather, these differences may be a result of deficiencies in the FLEXPART simulations, e.g., related to an under-representation of winter-time atmospheric inversion layers in the meteorological input data which would hinder vertical transport, or an underestimation of the observed winter deposition due to winter snow being blown away at the ELB site.*"

We now add a sentence in the conclusion on this point: "*The ice-core trends are less documented for winter than for summer. A better understanding of past ammonium changes in winter motivates the search for another glacier site in the Caucasus that possibly experiences a better preservation of winter snow (less wind erosion).*"

The manuscript is too long and not clear enough. The authors should adjust it so that the structure of the manuscript is expressed more clearly and concisely.

*We appreciate the reviewer's comment but would like to clarify that we are a bit uncertain about the suggestion to shorten or reorganize the manuscript. The other two reviewers did not recommend any structural changes. In fact one suggested expanding the discussion section while the other recommended publishing the manuscript as is. Additionally, the ACP journal does not provide explicit guidelines on manuscript length. However, in response to the feedback, we have reduced the number of figures in the main text, moving two to the Supplementary Information as recommended.*

Line 23, "0.5°x0.5°" should to be multiplication sign "×", not the letter "x". Note the change!
*Thank you for pointing this out. We acknowledge the convention of using the multiplication sign "×" instead of the letter "x". However, in our previous publications in JGR and GRL, as well as commonly in ACP papers, the "x" format (e.g., 5° x 5°) has been used. We will review this point carefully during the proof editing stage.*

Now that there are too many graphs in the manuscript, please organize some of them and place them in the supplementary Material.

*Thank you for your suggestion. While the journal does not impose limitations on the number of figures, we agree that streamlining the manuscript would be beneficial. In response, we have moved two figures to the Supplementary Information (SI): Figure 7 (Past changes of the agricultural area) and Figure 4b. Additionally, we have combined Figure 4a with Figure 9 to reduce redundancy and improve organization.*

---

## Author Comment (AC2)

**Reviewer 3.**

The authors investigate historical atmospheric ammonia (NH3) pollution using a 182 m long ice core from Mount Elbrus. The ice core data reveal a 3.5-fold increase in ammonium (NH4+) concentrations from approximately 1750 to 2009, with a significant rise post-1950 due to industrial and agricultural activities. The study utilizes FLEXPART atmospheric transport modeling to compare the ice core trends with past anthropogenic NH3 emissions, highlighting the substantial impact of human activities on atmospheric NH3 levels. The research also differentiates between natural and anthropogenic contributions to NH4+ concentrations, providing a baseline for pre-industrial natural emissions and underscoring the predominance of agricultural emissions in recent decades. The authors are leaders in this type of work; the data and methodology are all sound; and the topic and scope will be of interest to ACP readers. Overall, this article makes a significant contribution to the understanding of historical ammonia emissions in south-eastern Europe. Its robust dataset, interdisciplinary approach, and detailed methodology are commendable. I believe the paper is publishable mostly as is, but I encourage the authors to consider these points when revising:

Thank you for your thoughtful review.

**Model Assumptions and Limitations**: The study relies heavily on the FLEXPART model, which, while robust, has limitations. The assumption that atmospheric transport has not changed significantly over the ice core record period might oversimplify complex atmospheric dynamics. Some type of sensitivity analyses to explore the impact of varying transport conditions would be helpful. Thank you for your valuable feedback regarding the FLEXPART model and its limitations. We agree that the assumption of constant atmospheric transport over the ice core record may oversimplify the complexities of atmospheric dynamics. In response, we have now included an analysis of the temporal variability of emission sensitivities over the 1980-2019 period. As shown in the figure added to the Supplementary Information (see below), the emission sensitivities did not change significantly during this period.

[Figure]

Variability of summer emission sensitivity – Mt. Elbrus

**Winter Data Uncertainties**: The ice core data for winter months are less reliable due to fewer samples and potential wind erosion. This limitation weakens the study's conclusions about seasonal variations in NH3 emissions. Additional measures or methods to improve winter data accuracy would strengthen the overall findings. *Thank you for highlighting the limitations of the ice core data for winter months, including fewer samples and potential wind erosion. We agree that these factors can weaken the study's conclusions regarding seasonal variations in NH3 emissions.*

*While it is currently not feasible to implement additional measures to improve winter data accuracy, we have added the following statement in the conclusion: "The ice-core trends are less documented for winter than for summer. A better understanding of past ammonium changes in winter motivates the search for another glacier site in the Caucasus that may provide better preservation of winter snow (due to less wind erosion)." We believe this addition emphasizes the need for further investigation in this area.*

**Spatial Resolution of Emission Sources**: The study identifies significant contributors to NH4+ deposition, but the spatial resolution of these sources could be improved. A finer resolution might reveal more localized sources and patterns of emissions, offering better-targeted mitigation strategies. *We agree that a finer resolution could provide a more detailed understanding of localized sources and emission patterns, ultimately leading to better-targeted mitigation strategies. In this study, we utilized emissions data available at a resolution of 0.5° × 0.5°, which aligns with the resolution of the FLEXPART simulations. This resolution allows us to identify "hot spots" in several regions, such as Brittany in France, northern Italy, and western Ukraine (see Figure 2).*
*We appreciate your suggestion, as it highlights an important consideration for future research.*

**Consideration of Other Pollutants**: While the focus on NH3 is clear, the interplay between NH3 and other atmospheric pollutants (e.g., SO2, NO2) is mentioned but not deeply explored. A more detailed examination of how these pollutants interact and affect NH3 deposition could provide a fuller picture of atmospheric chemistry dynamics.

*We agree that a more detailed examination of these interactions could provide a fuller understanding of atmospheric chemistry dynamics. However, $NH_3$-$NO_x$-$SO_2$ interactions are quite complex and not always fully understood. As concluded in the paper, further progress in this area would require the deployment of a comprehensive transport-chemistry model that includes a complete description of $SO_2$, $NO_x$, and $NH_3$ chemistry, as well as the effects of historical dust changes. Unfortunately, such a model is not currently available.*

**Climate Change Implications**: The study briefly touches on the potential impact of climate change on NH3 emissions but does not delve deeply into future projections. Integrating climate models to predict future NH3 emissions under different climate scenarios would add valuable forward-looking insights.

*Thank you for your insightful comment. We agree that integrating climate models to predict future NH3 emissions under various climate scenarios would provide valuable insights. However, as with your previous comment, discussing future projections is challenging without the use of transport-chemistry model simulations that account for anticipated changes in SO2 and NO2 emissions, as well as dust aerosol.*

*In response to both your comments we have noted in the conclusion that "Transport-chemistry model simulations are welcome to further evaluate NH4+ ice core records; however, they would require consideration of increasing SO2 and NO emissions, as well as dust aerosol and its heterogeneous interactions with acidic species and NH3."*

---

## Author Comment (AC3)

**Reviewer 2.**

The manuscript presents a thorough analysis of ammonia (NH3) emissions over several centuries using an ice core record from Mount Elbrus. It focuses on both natural and anthropogenic sources of ammonia in the atmosphere and emphasizes the critical sources from agriculture. The study is significant as it provides insights into historical emission trends and their environmental impacts, contributing valuable data to atmospheric and environmental science, especially in the context of increasing agricultural practices and climate change. The proposed methodology and results in this paper are commendable and will undoubtedly serve as a reference point for future research in the field. Here are my specific suggestions.

*We would like to sincerely thank the reviewer for the thoughtful comments, which have helped us improve and clarify the manuscript.*

The introduction is generally clear, it might benefit from a brief overview of the significance of ammonia emissions in the context of atmospheric science and environmental policy to set a stronger foundation for the research.

*Thank you for the suggestion. We agree that highlighting the significance of ammonia emissions is important. In fact, we have already addressed this in the introduction, where we discussed the role of ammonia in atmospheric chemistry and its environmental impact: "Gaseous ammonia is the most abundant alkaline gas in the atmosphere and represents a major component of total reactive nitrogen. It plays an important role in determining the overall acidity (alkalinity) of precipitation. A large portion of atmospheric aerosols, acting as cloud condensation nuclei, consists of sulfate neutralized to various extents by $NH_3$. Ammonia and ammonium (collectively abbreviated as $NH_x$) are key nutrients that fertilize plants. Too large inputs of N to the environment may, however, lead to eutrophication of terrestrial and aquatic ecosystems and thus threaten the biodiversity (Asman et al., 1998; Galloway et al., 2003). Therefore, the growing $NH_3$ emissions resulting from fertilization applied to meet the need to sustain food production for a growing human population impact the environment."*
*We believe this provides a strong foundation for the significance of ammonia emissions in the context of atmospheric science and environmental policy.*

The methodology for the comparison process in this paper, including statistical methods or software tools employed, and any adjustments made for the comparison should be clearly outlined. Ensuring that the ice core data, model input files, and emission inventories are publicly available would greatly enhance reproducibility.
Providing access to these datasets through repositories or as supplementary materials is important. This can ensure that other researchers can replicate the findings reliably.

*Thank you for the suggestion. The methodologies used in this paper as recognized in your next comment are clearly outlined in the text. We did not apply specific statistical methods for the comparison process.*

*Regarding software tools, section 2.2 already details that past NH4+ deposition fluxes at the ELB site were calculated by weighting past NH3 emissions from each grid cell of the inventory by its emission sensitivity and summing over all grid cells to obtain the simulated deposition rate. These calculations were performed using the CDO (Climate Data Operator) software.*

*As for emission inventories, we used the global dataset of anthropogenic NH3 emissions (Hoesly et al., 2018), presented in NetCDF format with a 0.5° × 0.5° spatial resolution. This dataset is*

*publicly available (see Figure 2 caption, Hoesly et al., 2018: https://github.com/JGCRI/CEDS/), so there is no need to replicate it here.*

*We are unable to provide access to the meteorological input data used in the model simulations, as these datasets amount to hundreds of terabytes and we do not have permission from ECMWF to share them.*

*Lastly, as mentioned in the "Data Availability" section, ammonium concentration data can be accessed at https://zenodo.org/records/12549687 (Legrand et al., 2024).*

Although the methods are described in detail, some sections could use additional clarity, particularly for readers less familiar with specific techniques. It is recommended to supplement the text with more detailed explanations of these methodologies and any assumptions made during the analysis. Consider adding diagrams or flowcharts to visually represent the process.

Thank you for your valuable feedback. Although the methods are described in detail. Data discussed in this paper (mainly ammonium and sulfate) were obtained using the well-known ion chromatography. There is no specific assumption made behind such routine measurements. The reader can find detailed working conditions in earlier publications as referenced here (Preunkert and Legrand, 2013, Legrand et al., 2013). In the revised version, however, we have also included the following statement "*the detection limit for ammonium is close to 1 ng g$^{-1}$ so remaining well below mean (low) winter concentrations that typically ranged from 10 to 20 ng g$^{-1}$.*"

It is suggested to expand certain sections of the discussion to provide a more in-depth analysis of the implications of the findings, particularly expanding the discussion to cover the broader implications of the findings, potential limitations, and areas for future research would be beneficial.

*Thank you for your insightful suggestions. We recognize the importance of expanding certain sections of the discussion to provide a more comprehensive analysis of the implications of our findings. In the conclusion, we have already emphasized that "Transport-chemistry model simulations are welcome to further evaluate NH4+ ice core records; however, they would require consideration of increasing SO2 and NO emissions, as well as dust aerosol and its heterogeneous interactions with acidic species and NH3."*

*Additionally, we have now included the following statement: "The ice-core trends are less documented for winter than for summer. A better understanding of past ammonium changes in winter motivates the search for another glacier site in the Caucasus that may provide better preservation of winter snow (due to less wind erosion)." We believe these additions help to highlight potential areas for future research.*

---

## Author Response (AR2)

**Editor.**
The authors have addressed the reviewers' questions and concerns comprehensively. As required by the journal, however, the current abstract and concluding section have several missing elements that we are obliged to request the authors to add before the paper can be accepted. Please follow the guidelines provided in the link below to revise the abstract and concluding sections; after the editor's brief review of the final revision, the manuscript should be able to be accepted as it is.

We would like to sincerely thank the editor, which have helped us improve the manuscript.

1) The abstract has been reworded to meet all requirements, including the word count limit of fewer than 250 words.
2) In the conclusions the sentence: "That permitted to investigate ammonia pollution in south-eastern Europe, and that, well before the onset of the industrial period in 1850," – was reworded:
   *"With respect to previous studies conducted in the Alps in relation with pollution in western Europe since 1850, this study permitted to investigate ammonia pollution in south-eastern Europe, and that, well before the onset of the industrial period in 1850."*
3) We added a sentence in the conclusion:
   *"Overall, this work on historical $NH_3$ emission trends, provide valuable information to atmospheric and environmental science, especially in the context of increasing agricultural practices and climate change."*